# Switching imidazole reactivity by dynamic control of tautomer state in an allosteric foldamer

David P. Tilly [1,2] ✉, Jean-Paul Heeb [1], Simon J. Webb [2] & Jonathan Clayden [1] ✉

Molecular biology achieves control over complex reaction networks by means of molecular systems that translate a chemical input (such as ligand binding) into an orthogonal chemical output (such as acylation or phosphorylation). We present an artificial molecular translation device that converts a chemical input – the presence of chloride ions – into an unrelated chemical output: modulation of the reactivity of an imidazole moiety, both as a Brønsted base and as a nucleophile. The modulation of reactivity operates through the allosteric remote control of imidazole tautomer states. The reversible coordination of chloride to a urea binding site triggers a cascade of conformational changes in a chain of ethylene-bridged hydrogen-bonded ureas, switching the chain's global polarity, that in turn modulates the tautomeric equilibrium of a distal imidazole, and hence its reactivity. Switching reactivities of active sites by dynamically controlling their tautomer states is an untapped strategy for building functional molecular devices with allosteric enzyme-like properties.

In living systems, complex networks of signalling pathways translate physical or chemical signals into biochemical function by modulation of the conformational and binding properties of receptors and other biomolecules. Allosteric effects[1–3] not only transmit a signal through space to a location remote from the signalling input, but also translate one type of physicochemical signal into a chemically unrelated output[4–6]. Thus photoreceptor proteins[7] induce membrane depolarisation as a result of a light signal, and G protein-coupled receptors regulate the interconversion of primary metabolites as a response to the binding of structurally unrelated hormones or environmental signals[8,9].

Synthetic molecular translation devices that use allosteric changes to convert one type of chemical signal into an orthogonal output have been devised, in the form of dynamic foldamers[10–15] and other supramolecular systems[16–20]. Dynamic foldamers are extended oligomeric structures that display well-defined global conformational preferences, which may be modulated by specific stimuli[21] such as solvent polarity[22,23], ligand[24,25], light[26–28] and pH[29–31]. However, the application of dynamic foldamers so far has been limited by the nature of their

outputs, which have typically been restricted to stereochemical changes that produce spectroscopic (NMR or fluorescence) signals. Foldamers can also operate as catalysts[32,33], but dynamic foldamers that translate chemical signals into the regulation of reactions have been limited to the switching of product stereochemistry[31,34,35].

Imidazole is widely used in synthetic chemistry because of its Brønsted basicity and nucleophilic catalyst properties and is ubiquitous in biological systems[36]. Histidine plays an essential role in the 'catalytic triad' of active site residues in many enzymes, including trypsin, chymotrypsin, and acetylcholinesterase[37,38], where its imidazole acts as both a general acid and a general base, allowing protons to shuttle between substrate and catalyst. Both natural functions and synthetic uses of imidazole employ the lone pair of the unprotonated nitrogen to achieve reactivity—the other (protonated) nitrogen is unreactive as a base and a nucleophile. Yet the interchange between the different reactivities at each nitrogen atom—inert or basic/nucleophilic—is in principle simply a matter of tautomerism. We reasoned that the imidazole tautomer states could be switched by exchanging an intramolecular hydrogen-bond donor for an

[1]School of Chemistry, University of Bristol, Cantock's Close, Bristol BS8 1TS, UK. [2]Department of Chemistry, University of Manchester, Oxford Road, Manchester M13 9PL, UK. ✉e-mail: david.tilly@manchester.ac.uk; j.clayden@bristol.ac.uk

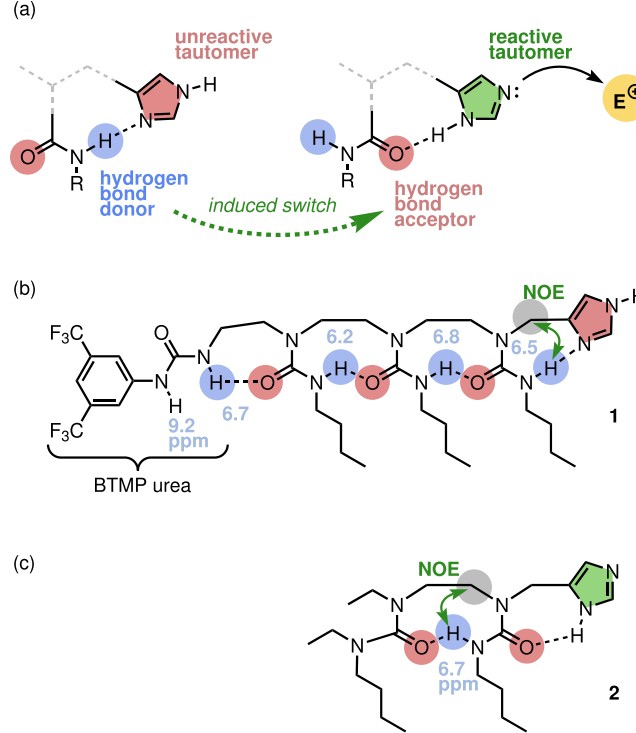

**Fig. 1 | Reactivity controlled by imidazole tautomrism. a** Design concept for an imidazole-based translation device. **b** The prototype device **1** with conformationally diagnostic chemical shifts (ppm) and NOEs. BTMP = *N*-3,5-bis(trifluoromethyl)phenyl. **c** 'Benchmark' foldamer **2**.

intramolecular hydrogen-bond acceptor[39]. Switching the polarity of the hydrogen bond to the proximal nitrogen atom would activate the imidazole by switching its tautomer state, incorporating the imidazole NH as a hydrogen bond donor, and revealing a reactive nitrogen sp[2] lone pair (Fig. 1a). We showed previously that dynamic foldamers built from ethylene-bridged oligomeric ureas display global switchable hydrogen-bond directionality[40–42], and we reasoned that this could provide the mechanism for the required induced polarity switch.

We now report a molecular translation device that allows a chemical signal to activate the reactivity of a remote imidazole active site by triggering a tautomer switch through a well-defined signal transduction mechanism involving a global reversal of hydrogen bond polarity. A true translation device requires polarity to be induced by a signal input that is itself neither a base nor a nucleophile; we made use of a chloride ion to perform this function.

## Results and discussion

A prototype device **1** (Fig. 1b) was designed and synthesized (Supplementary Fig. 1) by appending an imidazole to the terminus of an ethylene-bridged oligomeric urea. Similar oligomers have been shown to adopt a uniform global directionality with an uninterrupted linear chain of hydrogen bonds[40]. The structure of the hydrogen bond chain in **1** (7.4 mM in $CD_2Cl_2$ at 25 °C) was established by NMR spectroscopy (Supplementary Figs. 25–35). The [1]H NMR chemical shifts shown in Fig. 1b indicate that the urea NH groups are engaged in intramolecular hydrogen bonding, and a strong NOE correlation between the imidazolylmethylene protons and the adjacent urea NH confirms the directionality of the hydrogen-bond chain (Supplementary Fig. 29). Dilution of **1** in $CD_2Cl_2$ at 25 °C (from 118.2 mM to 7.4 mM) revealed that self-association was weak ($K_a = 31 \pm 2$ M[−1]). Taken together, these data are consistent with the adoption by the prototype device **1** of the conformation shown, in which the terminal *N*-3,5-bis(trifluoromethyl) phenyl (BTMP) urea acts as a hydrogen bond donor, and the basic

imidazole nitrogen atom acts as an intramolecular hydrogen-bond acceptor.

For comparison, imidazole-appended 'benchmark' foldamer **2** (Fig. 1c) was made (Supplementary Fig. 2), which has a tetrasubstituted terminal urea that cannot act as a hydrogen bond donor. Molecule **2** should adopt a hydrogen bond chain directionality opposite to the one displayed by **1**, with the imidazole switching from hydrogen bond acceptor to hydrogen bond donor. The contrasting conformation of **2** was confirmed by an NOE correlation between the hydrogen-bonded urea NH and the methylene protons of the ethylene bridge, and only a weak NOE correlation with the imidazolylmethylene protons (Supplementary Figs. 45, 46).

The opposite hydrogen-bond directionalities of **1** and **2** necessitate a switch in imidazole tautomer state. [13]C NMR spectroscopy was used to identify the tautomer preferences of **1** and **2**. First, using 4- and 5-methyl-1*H*-imidazole models, we calculated by DFT (GIAO, B3LYP/6-311 G(d) and ωB97-XD/6-311 G(d)) the [13]C NMR chemical shifts of the ring carbon atoms in these two alternative tautomer states (Fig. 2a, b). These calculations revealed that the [13]C NMR chemical shift values of C4 and C5 vary substantially upon tautomerisation and that the difference in [13]C NMR chemical shifts between C4 and C5 are diagnostic of the tautomer state. C4 and C5 differ in calculated chemical shift by about 30 ppm in the 4-alkyl-1*H* tautomer (red), but have almost identical calculated shifts in the 5-alkyl-1*H* tautomer (green). Experimental support for the use of chemical shift separation between the [13]C signals of C4 and C5 as a diagnostic indicator of tautomer state was provided by the empirical [13]C NMR chemical shifts values of C4 and C5 for equivalent regioisomeric compounds 1,4-dimethylimidazole and 1,5-dimethylimidazole, in which an *N*-methyl group replaces the tautomeric NH (Fig. 2c)[43].

Tautomers undergo rapid exchange on the NMR timescale, and low temperature experiments were hampered by insolubility. The experimental [13]C NMR chemical shift values of C4 and C5 in the imidazole ring of **1** and **2** are, therefore, weighted averages of the two tautomer states, but the change in the separation of these two peaks indicates that the tautomer states are differently populated in **1** and **2** (Fig. 2d). The difference in chemical shift between C4 and C5 of **1** (δC4−δC5 = 21 ppm, Supplementary Fig. 32) is consistent with a major imidazole tautomer that allows an intramolecular hydrogen bond between a urea NH and the imidazole's basic nitrogen atom. The smaller difference in **2** (δC4 − δC5 = 11 ppm, Supplementary Figs. 49–52) indicates a greater population of the alternative tautomer, in which an intramolecular hydrogen bond is able to form between the adjacent urea C = O and the imidazole NH. [1]H NMR chemical shifts by contrast turned out not to be diagnostically useful: the corresponding imidazole CH protons in **1** and **2** are almost identical in chemical shift.

The influence of induced tautomer preference on the reactivity of each ligated imidazole was first assayed by comparing the basicity of **1** and **2** (Fig. 3). In a preliminary experiment, a solution of 2,4-dinitrophenol (1.16 mM in $CD_2Cl_2$) was titrated with 0–12 equiv. imidazole (Supplementary Fig. 82). The solution visibly changed from colourless to yellow, characteristic of the 2,4-dinitrophenolate anion, as the base was added. During the titration the absorbance decreased between 250–300 nm and increased between 330–440 nm, with an apparent isosbestic point at 315 nm (Fig. 3a); no further change was observed after 10 equiv. imidazole had been added. Neither imidazole, nor foldamer **2**, absorbs at wavelengths between 250–520 nm at the concentrations used for the titration, and no precipitate was formed during the titrations. The final spectrum matches that of tetrabutylammonium 2,4-dinitrophenolate (Supplementary Fig. 83). Overall, the titration data are consistent with the progressive deprotonation of 2,4-dinitrophenol [p$K_a$(H$_2$O) = 4.0] by imidazole [p$K_a$(H$_2$O) = 6.95] to form imidazolium 2,4-dinitrophenolate[44].

A similar experiment was conducted with 0–12 equiv. of benchmark foldamer **2**, with very similar results (Fig. 3b, Supplementary

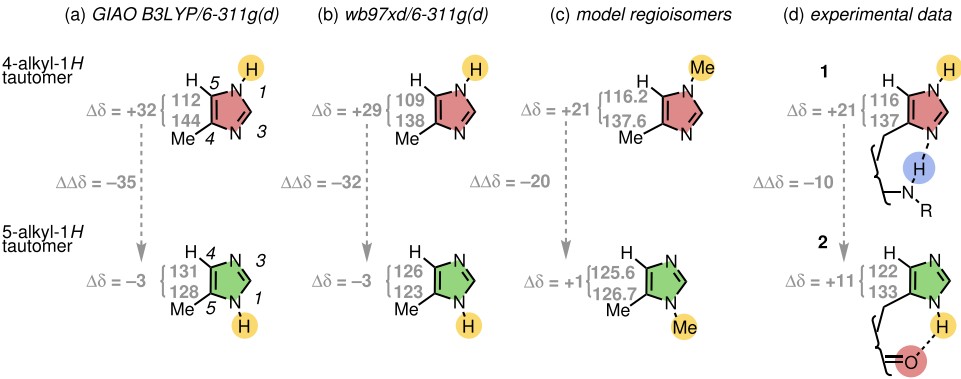

**Fig. 2 | Identifying the tautomer state of a substituted imidazole by ¹³C NMR spectroscopy. a, b** Computed ¹³C NMR chemical shifts (using two different functionals, Supplementary Fig. 55) for the two tautomers of methylimidazole (Δδ values shown in ppm). **c** Model regioisomers 1,4-dimethylimidazole and 1,5-dimethylimidazole. **d** Experimental ¹³C NMR chemical shifts (CD₂Cl₂, 25 °C) for

device **1** and benchmark structure **2**. The difference in chemical shift (ΔΔδ in ppm) between C4 and C5 is consistently reduced significantly in the tautomer or regioisomer with the protonated or methylated nitrogen adjacent to the C-alkyl substituent.

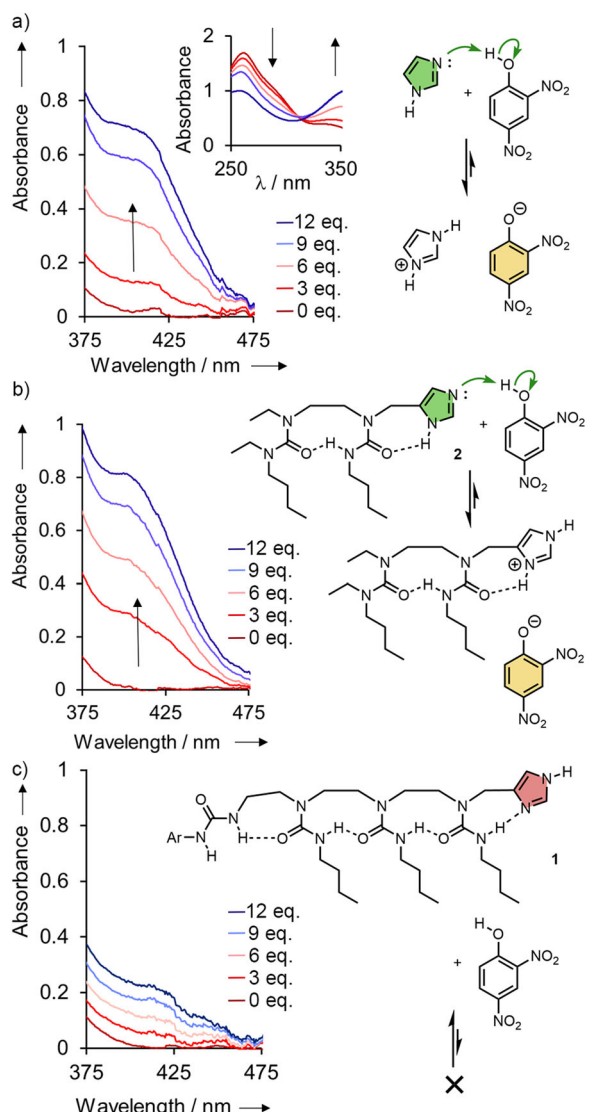

**Fig. 3 | Induced conformational effects on basicity.** Stacked UV-visible spectra acquired from the titration at ambient temperature of a solution of 2,4-dinitrophenol (1.1 mM in dry dichloromethane) with (**a**) imidazole, (**b**) benchmark foldamer **2** and (**c**) device **1**. In each case 0–12 equiv of the additive was used, in 1 equiv increments.

Fig. 87). A yellow colour accompanied the development of a spectrum characteristic of the 2,4-dinitrophenolate anion, again indicating deprotonation of 2,4-dinitrophenol by **2** to form a soluble 2,4-dinitrophenolate salt.

Next, we titrated under the same conditions a solution of 2,4-dinitrophenol with 0–12 equiv. of **1** (1 equiv. increments, Fig. 3c). This time, there was no colour change during the titration and no strong absorbance between 350–390 nm and 400–450 nm. There were no isosbestic points and progressive small increases in absorbance at all wavelengths between 220–450 nm, which are attributed to the absorbance of added **1** only (Supplementary Fig. 85): **1** does not deprotonate 2,4-dinitrophenol to any significant extent.

It, therefore, appears that opposite polarities in the ligated chain of hydrogen bonded oligoureas alter the reactivity of the imidazole terminal substituent. Compound **2**, whose tautomer state presents a terminal nitrogen lone pair, acts as a base towards 2,4-dinitrophenol whereas the predominant imidazole tautomer state in **1**, which presents only a terminal N–H, with the imidazole lone pair engaged in an intramolecular hydrogen bond, is less reactive. The difference in the tautomer state between **1** and **2** is induced by the availability of a remote NH at the opposite terminus of the chain. This prompts the possibility that an intermolecular hydrogen bond to this remote urea NH by a non-basic, non-nucleophilic ligand might likewise a induce a change in tautomer state, 'translating' the presence of the ligand into induced chemical function at the distant imidazole site.

Preliminary work to establish the conformational consequence of binding non-basic anions to a BTMP urea was carried out using the analogous receptor **3** (Fig. 4a, Supplementary Fig. 57) that has phenyl in the place of imidazole. The ¹H NMR chemical shifts of the internal urea NH groups of **3** (highlighted in blue in Fig. 4a) indicate that they are both intramolecularly hydrogen-bonded to adjacent carbonyl groups, while the terminal NH (at 4.44 ppm, in mauve), which shows an NOE correlation with the benzylic methylene group (Supplementary Figs. 61 and 63), is not. Titration of **3** (11.9 mM in CD₂Cl₂ at 25 °C) with tetrabutylammonium chloride (Bu₄NCl, 0 to 2.69 equivalents, Fig. 4b) led to progressive downfield shifts in the ¹H NMR signals of both NH signals of the BTMP urea (NH$_A$ and NH$_B$, Fig. 4a). More than 1 equivalent of ligand had minimal further effect (Fig. 4b, Supplementary Figs. 65–70), consistent with reversible 1:1 coordination of chloride to the BTMP urea terminus of **3**. Electrospray mass spectrometry revealed a 1:1 complex of **3** with Cl⁻, and non-linear curve fitting of the titration curves to a 1:1 binding model gave $K_a = 300 \pm 100$ M⁻¹ in CD₂Cl₂ (Supplementary Figs. 68 and 70).

Concomitant with the formation of this complex, the signal arising from the terminal, non-hydrogen-bonded NH (NH$_D$, Fig. 4a,

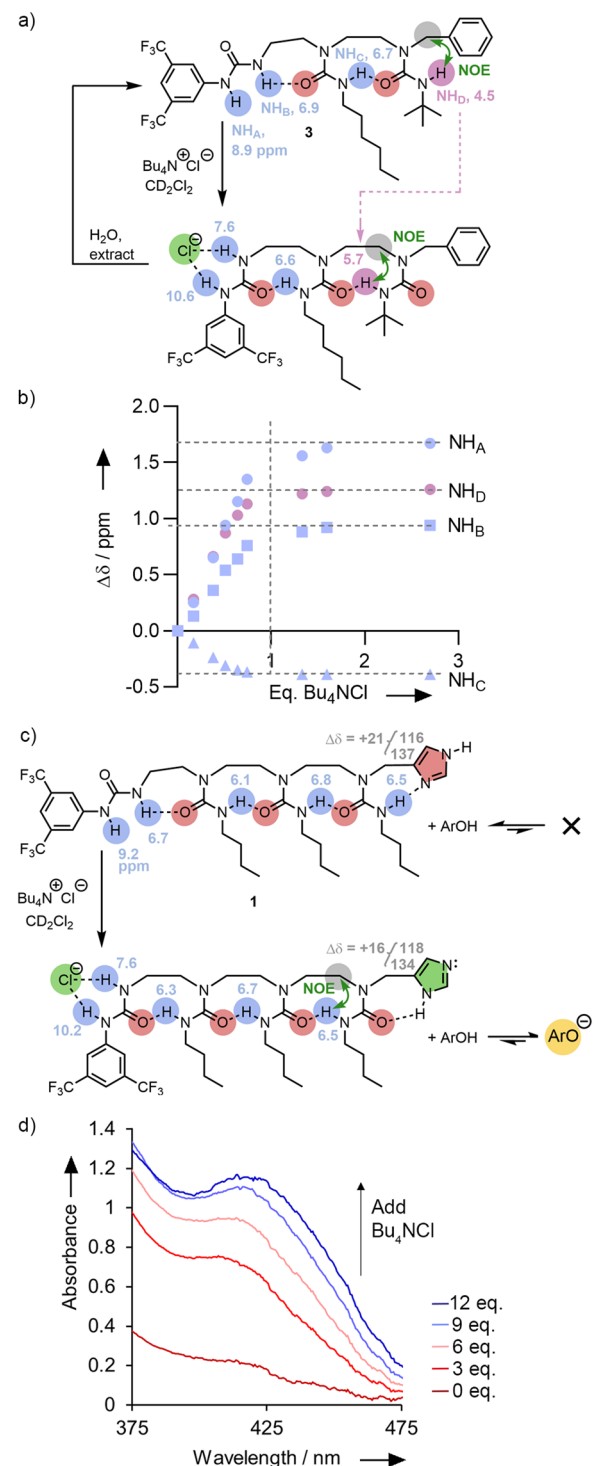

**Fig. 4 | Conformational changes revealed by NMR.** The conformational consequences of adding $Bu_4NCl$ (**a**) to foldamer **3**, showing (**b**) the change in $^1H$ NMR chemical shift ($CD_2Cl_2$) of the hydrogen-bonded NH groups on addition of 2.7 equiv $Bu_4NCl$, and (**c**) to device **1** with diagnostic changes in the $^1H$ and $^{13}C$ NMR spectra revealing a consequent change in tautomer state. **d** The effect on the UV-visible spectrum of adding $Bu_4NCl$ to a mixture of 2,4-dinitrophenol and **1** (12 equiv.) revealing the induced activation of basicity in the terminal imidazole.

only the trisubstituted ureas that form the hydrogen-bond chain of **3**[40], and on compound **2** (Supplementary Fig. 56) ruled out the possibility that the chemical shift variations measured during the titration arise from a direct interaction between $Bu_4NCl$ and the hydrogen-bond chain. Titration of bis(butylurea) **SI5** in $CD_2Cl_2$ at 25 °C with $Bu_4NCl$ (Supplementary Fig. 80) led to complexation-induced shifts (CIS) of the ureido NH $^1H$ NMR signals (CIS = + 0.08 ppm) significantly smaller than those recorded during the titration of foldamer **3** with similar amounts of ligand (CIS + 0.8 to +1.5 ppm). The titration of the longer tris(butylurea) **SI6** in $CD_2Cl_2$ at 25 °C with $Bu_4NCl$ also led to much lower complexation induced shifts (+0.23 ppm at 1.1 equivalents of ligand added) (Supplementary Fig. 81). We conclude that 1:1 coordination of chloride to the BTMP urea is translated into a reversible switch in the global hydrogen-bond directionality of foldamer **3**.

Other anions induce a similar change in conformation. Comparable binding studies of **3** with $Bu_4NBr$, $Bu_4NI$ and $Bu_4N(NO_3)$ (Supplementary Figs. 71–79) revealed changes in the $^1H$ NMR spectra analogous to those seen during the titration of **3** with chloride, albeit with smaller complexation-induced shifts in the BTMP urea NH protons (CIS of $NH_A$ and $NH_B$ respectively: $Br^-$ +1.2 and +0.7; for $I^-$, + 0.5 and +0.3 ppm). Fitting of the changes in NH chemical shift gives affinities for bromide and iodide that are similar to each other ($K_a(Br^-) = 170 \pm 20\ M^{-1}$; $K_a(I^-) = 150 \pm 20\ M^{-1}$) and around half the value for chloride. $^1H$ NMR titration of **3** with $Bu_4N(NO_3)$ also gave a clear CIS of +0.9 ppm in the BTMP urea NH protons. Fitting of these changes gave $K_a(NO_3^-) = 1400 \pm 600\ M^{-1}$, greater than the value for chloride despite the smaller CIS. These binding constants approximate the reported relative affinities for ureas in organic solvents of all anions tested, albeit with relatively stronger binding to nitrate[45,46]. The strong downfield shift in $NH_D$ during the titration of **3** with each of bromide, iodide or nitrate is consistent with a switch of hydrogen bond directionality bringing the terminal NH into an internally hydrogen-bonded environment.

Importantly, chloride ions induced similar effects with the device **1** (Fig. 4c). Titration with $Bu_4NCl$ (0 to 2 equiv., Fig. 4b, Supplementary Figs. 36–42) led to strong downfield shifts of +1.7 and +0.9 ppm of the $^1H$ NMR signals of the BTMP urea, indicating regioselective binding of the chloride at the BTMP site. One equivalent of chloride was enough to reach the maximum induced shift; minimal changes occurred upon further addition. The data fitted a 1:1 binding model with $K_a(Cl^-) = 1600 \pm 600\ M^{-1}$. Applying a 1:2 binding model also gave an adequate fit but showed that the amount of higher order complexation would be small (for $1·Cl^- + Cl^-$ $K_{12}(Cl^-) = 63 \pm 8\ M^{-1}$; Supplementary Fig. 39). The relatively small changes in the $^1H$ NMR chemical shifts (CIS < 0.2 ppm) of the internal NH groups (which remain hydrogen bonded in both conformers) do not give diagnostic conformational information. The small upfield and downfield movements in the positions of these NH resonances during the titration could not be analysed quantitatively, but may result from the effect of chloride-induced changes in self-association of **1**: dilution studies had shown that **1** weakly self-associates in $CD_2Cl_2$ ($K_{assoc} = 31 \pm 2\ M^{-1}$), producing small downfield shifts in all NHs (Supplementary Information Section 5.1.2).

Crucially, the addition of chloride leads to a clear increase in the NOE correlation between the terminal urea NH and the ethylene chain (Fig. 4c, Supplementary Fig. 40), indicating an increased population of the new conformer. A concomitant change in the tautomer state of the imidazole was indicated by the diagnostic change in the C4 and C5 $^{13}C$ NMR chemical shift values of **1** in the presence of 2 equiv. $Bu_4NCl$ in $CD_2Cl_2$ ($\delta C4 - \delta C5 = 15.8$ ppm, Fig. 4c, Supplementary Figs. 31, 41). It was not possible to observe the imidazole NH by $^1H$ NMR spectroscopy, even at low temperature, due to exchange broadening.

The chloride-induced change in the tautomer state of **1** provides the basis for the operation of a molecular translation device, in which an input signal – the presence of chloride – induces remote and unrelated chemical effects. To demonstrate this, $Bu_4NCl$ (0–12 equiv.)

Supplementary Figs. 61, 63, 69) migrated significantly downfield during the addition of the first equiv. of $Bu_4NCl$, and developed an NOE to the adjacent ethylene bridge. A simple aqueous wash restored all the $^1H$ NMR signals of **3** to their original shifts. Control experiments on oligoureas **SI5** and **SI6** (Supplementary Figs. 80, 81), which contain

was titrated into **1** (12 equiv.) in the presence of 2,4-dinitrophenol (1.1 mM in dry dichloromethane). As chloride was added, a yellow colour developed, alongside a progressive increase in absorbance between 330–470 nm (Fig. 4d). An isosbestic point was observed at 310 nm as Bu₄NCl alone shows no significant absorbance in this region. The UV spectra of neither 2,4-dinitrophenol nor **1** is changed by addition of tetrabutylammonium chloride (Supplementary Figs. 86, 91). Overall, the titration spectra are consistent with the progressive deprotonation of 2,4-dinitrophenol caused by increased formation of a complex of **1** and Cl⁻. They contrast sharply with the lack of basicity in **1** alone. These results mirror the way in which basicity depends upon tautomer state in chloride-free **1** and benchmark structure **2**.

The chloride-induced basicity of **1**, in conjunction with the regioselective binding and switch of directionality of its hydrogen bond chain, implies that chloride is capable of 'priming' the reactivity of the imidazole; switching its tautomer state from one that presents an unreactive NH group to one that presents a basic lone pair which is also nucleophilic[47]. The consequence of this induced tautomer switch on the nucleophilicity of the imidazole was also explored, by presenting the oligourea with an electrophilic partner, bis(4-nitrophenyl) carbonate **4** (Fig. 5). A preliminary experiment showed that electrophile **4** (65.6 mM in dry CD₂Cl₂ at ambient temperature) reacts with imidazole (1 or 3 equiv.) over a period of 13 h to give 4-nitrophenol and 4-nitrophenyl 1H-imidazole-1-carboxylate **5** as the sole products (Fig. 5a, R = H, Supplementary Information Sections 6.1, 6.2), which were characterised by NMR spectroscopy and HRMS (Supplementary Figs. 100–108). An excess of imidazole does not react further with **5**, nor does it measurably deprotonate 4-nitrophenol ($pK_a(H_2O) = 7.07$) under these conditions. UV-visible spectroscopy of the reaction showed a progressive increase in absorbance between 300–370 nm, and a decrease between 250–280 nm (Supplementary Figs. 95, 110, 111).

To explore the ability of **1** to translate the presence of chloride ions into nucleophilic reactivity, **4** (0.37 mM in CD₂Cl₂ at ambient temperature) was treated with 3 equiv. of either imidazole, **2**, **1**, or **1** + Bu₄NCl, and the reactions were monitored at 327 nm, the absorbance maximum of the respective acylated imidazole adducts (Fig. 5c, Supplementary Information Section 7). In the presence of imidazole or **2**, (Fig. 5a), the absorbance steadily increased over 600 min, with 4-nitrophenol and **5** (from imidazole) or **6** (from **2**) being the sole products by ¹H and ¹³C NMR spectroscopy and by HRMS (Supplementary Figs. 112–116). With **1** alone, the absorbance remained virtually unchanged over 13 h, and ¹H NMR spectroscopy showed only traces of the adduct **7**. Device **1** is evidently not only less basic than **2** or imidazole itself but also less nucleophilic.

When the experiment with **1** was repeated in the presence of Bu₄NCl (Fig. 5b, green trace), the absorbance at 327 nm increased over 600 min. (Fig. 5c), in sharp contrast with the data recorded for **1** in the absence of Bu₄NCl. The HRMS of the reaction mixture also confirmed the formation of the adduct **7** (Supplementary Fig. 118). A series of control experiments were carried out (Supplementary Figs. 119–124) to rule out alternative explanations for these observations: in summary, bis(4-nitrophenyl)carbonate did not react with an analogue of **1** lacking the imidazole terminus and Bu₄NCl, or with Bu₄NCl or other tetrabutylammonium salts alone.

The data from each set of conditions were fitted to pseudo-first order kinetics to provide approximate relative reaction rates (Fig. 5c). This analysis showed a clear difference in reactivity between **1** and covalently activated **2**, and the binding of **1** to chloride provided an approximately 20-fold increase in the nucleophilic reactivity of the foldamer **1** ($k = 260 \times 10^{-6}$ s⁻¹). Data fitting shows the reaction of **4** with **1** in the presence of Bu₄NCl is two-fold faster than reaction of **4** with **2**, a foldamer with the imidazole covalently activated. Although this

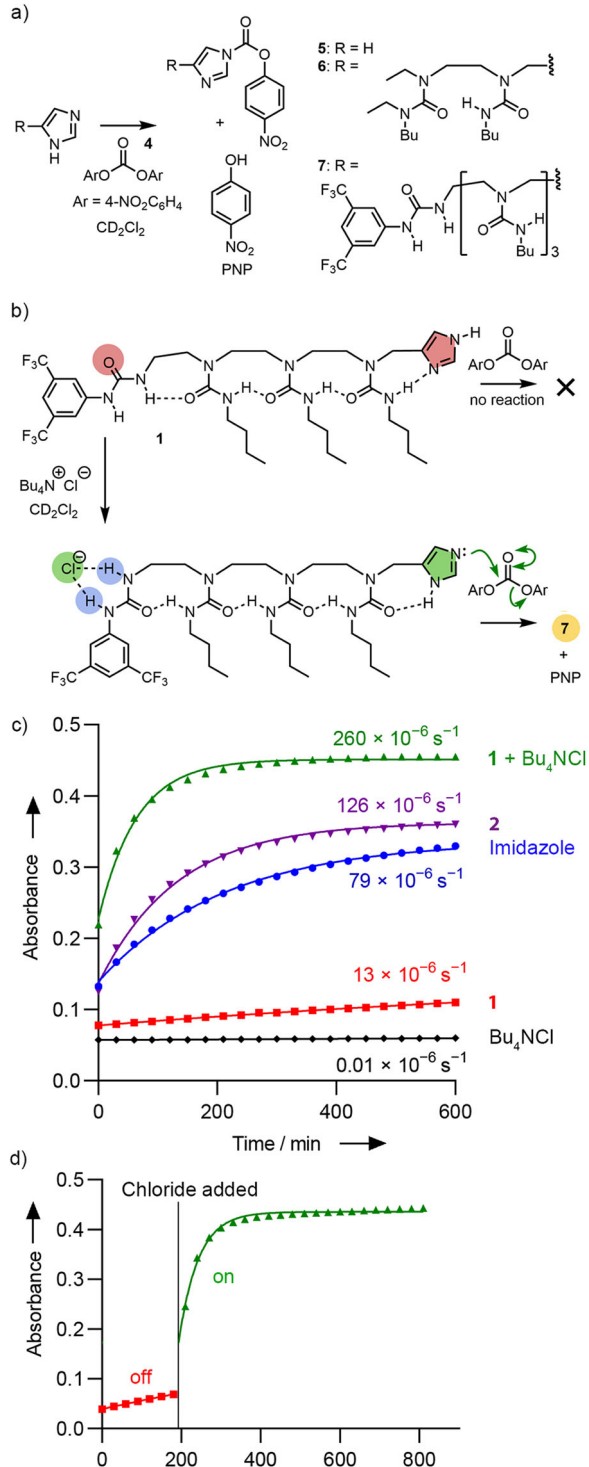

**Fig. 5 | Chloride-induced activation of a responsive foldamer.** The reaction of bis(4-nitrophenylcarbonate) **4** with (**a**) imidazole (R = H) to give adduct **5**; benchmark compound **2** to give adduct **6**; device **1** to give adduct **7**. **b** Activation of device **1** for reaction with **4** takes place only when **1** is bound to chloride. **c** Progress of the reaction of **4** with imidazole (in blue), **2** (in mauve), **1** (in red), **1** + Bu₄NCl (in green), and Bu₄NCl (in black). The increase in absorbance at 327 nm, arising from the adduct **7**, was monitored. The data are fitted to first-order kinetics to provide approximate observed rate constants (shown). **d** Switch-on of nucleophilic activity of **1** by the addition of chloride (3 equiv.) added after 193 min.

could be due to greater hydrogen bond polarisation in the longer foldamer[48], the reaction of **4** with **2** in the presence of Bu$_4$NCl is 1.4 times faster than with **2** alone (Supplementary Fig. 125). One possible explanation for this small enhancement is an increase in solvent polarity effects upon Bu$_4$NCl addition. Since other anions were able to switch the conformation of **3** then, like chloride, these other anions may also activate **1** as a nucleophile. An increase in the reactivity of **1** towards **4** was observed upon the addition of bromide or iodide, with fitting of the data providing $k = 130 \times 10^{-6}$ s$^{-1}$ for bromide and $57 \times 10^{-6}$ s$^{-1}$ for iodide. These rate accelerations mirror the relative affinity of **3** for each halide (I$^-$ < Br$^-$ < Cl$^-$). Despite binding more tightly to **3** than chloride, adding nitrate to **1** did not produce greater rate acceleration ($k = 140 \times 10^{-6}$ s$^{-1}$) than chloride addition, although it was greater than that of the other halide anions tested. A control experiment using **1** with Bu$_4$N(BPh$_4$), which has a non-coordinating anion, produced a small increase in rate compared to **1** alone ($k = 26 \times 10^{-6}$ s$^{-1}$ and $13 \times 10^{-6}$ s$^{-1}$, respectively) perhaps due to an increase in solvent polarity effects.

The magnitude of the Bu$_4$NCl induced increase in reactivity of **1**, together with NMR data indicating the stoichiometric binding of chloride to **1** at the BTMP site produces a pronounced polarity switch, clearly indicates that the allosteric effect of chloride on the tautomer state of the imidazole is the major contributor to the overall rate increase. This switch-on of activity is clear upon delayed addition of Bu$_4$NCl to the slow reaction of **1** with **4**, which resulted in an immediate rate acceleration (Fig. 5d). The relative rates of reaction in both regions of this time course could be approximated by fitting the data to pseudo-first order kinetics. The off-state before chloride addition gave $k \sim 12 \times 10^{-6}$ s$^{-1}$, whereas the on-state after chloride addition showed a strong increase in the observed rate constant, calculated as $k_{Cl} \sim 340 \times 10^{-6}$ s$^{-1}$. These values are in reasonable agreement with those calculated separately for each set of conditions (Fig. 4c).

In conclusion, we have built a dynamic molecular device with allosteric enzyme-like properties. It operates by an original mechanism in which remotely induced switching between the tautomer states of an active site modifies reactivity. The binding of chloride reorganises the hydrogen-bonding network of **1**, inducing a shift in the tautomeric state of the imidazole terminus from an off state to an on state, activating the device as a Brønsted base and as a nucleophile. The low basicity and nucleophilicity of chloride reveals the ability of foldamer **1** to act as an artificial translation device, converting one form of chemical input into a functionally orthogonal chemical output. Modulating reactivities of active sites through the dynamic control of their tautomer states is an untapped strategy to build functional molecular devices. Future work will seek to integrate such components as control devices in more complex reaction networks.

## Methods

The Supplementary Information provides details of methods for the synthesis of oligomers **1**–**3**, their spectroscopic characterisation and conformational analysis, their titration with anions, and kinetic studies of their reactions.

## Data availability

The data generated in this study are provided in the Supplementary Information. Source data are provided with this paper. All other data are available from the corresponding authors upon request. Source data are provided with this paper.

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

## Acknowledgements

The work was supported by the EPSRC through Programme Grant *Molecular Robotics* EP/P027067/1 (J.C and S.W.) and the Bristol EPSRC Centre for Doctoral Training in Technology-Enhanced Chemical Synthesis EP/S024107/1 (studentship to J.P.H.), the European Commission through a Marie Sklodowska Curie fellowship REFOLDAMER (DPT), and by the ERC through Advanced Grant DOGMATRON 883786 (J.C.). We thank Prof. Craig Butts for valuable discussions.

## Author contributions

J.C. and D.P.T. devised the project and designed the molecular structures. D.P.T., S.J.W. and J.C. planned the experiments and analysed the data. D.P.T. carried out the experimental work. J.P.H. carried out the computational work. All authors wrote the manuscript.

## Competing interests

The authors declare no competing interests.
