## [Peer Review File · Nature Communications]

Switching imidazole reactivity by dynamic control of tautomer state in an allosteric foldamerReviewers' Comments:

Reviewer #1:

Remarks to the Author:

The authors report a urea oligomer with conformational control of reactivity at an imidazole. Chloride binding at a distal urea induces conformation switching and activates nucleophilic reactivity of the imidazole. Control experiments establish that reactivity switching is unique to the urea oligomer and no reaction occurs in the presence of chloride alone. The authors have corroborated their findings with control molecules and extensive spectroscopic characterization. The identity of new molecules is well supported with characterization data in the Supporting Information. Foldamers with controlled switching are an active area of research and the results presented are novel within the field. This work is highly relevant to anyone working in this field and would be of high impact for others interested in controlled reactivity.

The authors have established that chloride activates imidazole and a non-coordinating anion, BF_4^- , does not. Other coordinating anions have not been addressed in this work, such as iodide, bromide, or nitrate. These anions compete with chloride binding to ureas, and this leaves open the question whether activation is specific to chloride. Revisions may be necessary to address this limitation when describing the oligomer as specific to chloride on line 181.

I recommend this article is accepted with minor revisions based on the impact of the work and strength of the supporting experiments.

Minor Corrections:

Figure SI93 – The green plot is not included in the legend for this graph.

Reviewer #2:

Remarks to the Author:

The paper by Tilly and others describes a polyamide foldamer presenting an imidazole in one side. After addition of a chloride, the reactivity of the imidazole moiety is enhanced. While the paper presents a large amount of data and the results are intriguing, I am not convinced by the origin of this effect and more studies should be carried out.

In particular, I refer to three major issues:

- 1) Imidazole proton (number 8 in Figure SI 12) is not changing chemical shift after the addition of chloride. I did not find any explanation/suggestion for this. In my opinion the chemical shift of that proton should be very sensible. I guess that the authors were expecting a variation. And if this is not the case a comment should be made.
- 2) Chemical shifts in Figure SI13 are clearly describing a process more complicated than a 1:1 stoichiometry. From dimerization data it does seem related to aggregation processes. However, chemical shift variations are not linear (e.g. NH_4 going first up and then down) indicating a more complex recognition pattern. This should be clarified (see also errors in the binding fitting).
- 3) How the authors are sure that the halide is not binding within the foldamer amides and only in the external part? This point needs also to be clarified.

The overall effect is interesting, but all these issues need to be clarified before publication.

Reviewer #3:

Remarks to the Author:

Tilly, Clayden and co-workers report on the design of dynamic foldamers for the allosteric control of the tautomeric state of imidazole, that translates into an on-off control of the nucleophilicity and basicity of this imidazole group. This work is part of a recently-developed strategy by the same group of the control of reactivity at one site of the foldamer by structural modification occurring at the opposite remote site of the foldamer. The three recently published papers (refs 36-38) somewhat rely

on the same concept, i.e. inversion of the polarity of the hydrogen bond chains in the foldamers, but the final outcome is clearly different, since it relies here on the reactivity of an imidazole ring by tuning its tautomeric state whilst previous papers rely on reversible binding on an anion and of an imide connected to the foldamer (promote by acid/base chemical inputs). Even though the outcome is not impressive (a 20-fold difference between the on and off states for the acylation of the imidazole), the concept appears original and seems plausible in regards of the presented data, notably the control experiments. On my point of view, the most striking point might be the selective bonding of the chloride at the extremity of the foldamer when considering the number of potential hydrogen bond acceptors in the foldamer. The following points can help clarifying certain aspects of the manuscript.

(1) Synthesis of ethylene-bridged oligoureas: it is not clear to me whether the synthesis scheme presented in the SI includes common intermediates with analogues published in refs 36-38. I wonder whether 1c, for example, is new or already used in previous studies.

(2) Tautomerization rate: population of imidazole tautomers lead to two separate peaks in ^{13}C NMR instead of four which means that the exchange is rapid on the ^{13}C NMR timescale. Is it possible to have a more quantitative assessment of the ratio of 4- and 5-isomers?

(3) Probing of the bonding of chloride: interpretation of the N-H shifts of 1 upon chloride binding are not obvious. Why NHC appears upfield shifted? Why the imidazole N-H is not mentioned as potential probe to follow this binding? FT-IR would have been a particularly suitable to follow this experiment in solution. The authors should also consider that other stoichiometries are possible (2:2, etc..) and that minor species with chloride at other positions cannot be excluded.

(4) Difference in reactivity between 2 and 2+chloride: The small effect of chloride on the activity of 2, which makes 1+Cl and 2+Cl of similar reactivity, is explained by "increase in solvent polarity". However, such an effect is not exhibited by Bu_4NCl alone. The authors must precise their interpretation.

Finally, a more precise background of the work will be given by adding existing literature on the effect of chloride on tuning the selectivity of helical catalysts (Chem. Commun., 2019, 2162) as well as quoting references dealing with switchable catalysts from the Schmittel group. I also encourage the authors to organize better their SI file (130 pages) which is quite hard to follow.

Detailed point-by-point response to referees' comments.

Reviewer 1

The authors have established that chloride activates imidazole and a non-coordinating anion, BF₄⁻, does not.

Other coordinating anions have not been addressed in this work, such as iodide, bromide, or nitrate. These anions compete with chloride binding to ureas, and this leaves open the question whether activation is specific to chloride. Revisions may be necessary to address this limitation when describing the oligomer as specific to chloride on line 181.

In response to this suggestion we have now carried out titrations of compound **3** with tetrabutylammonium bromide, iodide, and nitrate to evaluate their binding constants and the complexation-induced ¹H NMR shifts in their NH signals; we have compared these values with those for tetrabutylammonium chloride. These studies are reported in the SI, Sections 5.3.4-5.3.6 and in Figures S47-S55, and discussed in the manuscript in a new paragraph on p7.

We have also carried out reactions of bis(4-nitrophenyl)carbonate with compound **1** in the presence of each of tetrabutylammonium bromide, iodide, and nitrate, monitoring the reactions by time-course UV-vis studies (Figure S103) in order to estimate the effect of these salts on reaction rates (Table S9) in comparison with tetrabutylammonium chloride. Control experiments in which tetrabutylammonium bromide, iodide, and nitrate were mixed with bis(4-nitrophenyl)carbonate alone were also carried out (Figures S102, S66). These results are discussed in additional text on p10.

The results indicate that, like chloride, these anions do indeed also bind to, and induce a conformational change in, dynamic foldamer **3**. Tetrabutylammonium chloride exhibits higher binding affinity to compound **3** than the other halide anions, but it binds more weakly than nitrate. We have therefore removed the word 'selective' from the text as this could be misconstrued as meaning 'selective for chloride' rather than 'selective for binding at the BTMP urea'. The new text summarises the relative binding affinity of the full set of anions, which is consistent with data reported in the literature (now cited as reference 45). Interestingly, the change in chemical shift on adding chloride is greater than the other halides, as expected, but also greater than that induced by adding nitrate.

The rate data indicates that compound **1** reacts with bis(4-nitrophenyl)carbonate faster in the presence of chloride than with bromide, iodide, or nitrate, but that these other coordinating anions do exhibit some activating effect, as expected. All anions increase rate of reaction, with some correlation among the halides between the anion-binding affinity of the foldamer and the reaction rates. These observations are collated in the Table S10 in the ESI, and discussed in the new text on p10.

Minor Correction: Figure SI93 – The green plot is not included in the legend for this graph.

This oversight has been corrected: the data shown in has been added to this figure, which is now Figure S99 of the SI.

Reviewer 2

1) Imidazole proton (number 8 in Figure SI 12) is not changing chemical shift after the addition of chloride. I did not find any explanation/suggestion for this. In my opinion the chemical shift of that proton should be very sensible. I guess that the authors were expecting a variation. And if this is not the case a comment should be made.

We were indeed hoping that ¹H NMR chemical shifts around the imidazole ring would be diagnostic of tautomer state, but unfortunately the experimental data indicates this is not the case. For example, compounds **1** and **2**, show clear differences in ¹³C NMR data for the diagnostic signals of the imidazole ring, leading us to conclude that they adopt different major tautomer states. Their hydrogen-bond directionality is confirmed by NOE, and they have substantially different reactivity at the imidazole site. Despite these observations, **1** and **2** have ¹H chemical shifts for each of H8 and H9 that differ by only 0.06 ppm between the two compounds:

1: 7.60 (s, 1H, NCH^δN), 6.96 (s, 1H, CH^δ)

2: 7.56 (d, $J = 1.0$ Hz, 1H, $\text{NCH}^{\beta}\text{N}_{\text{imid}}$), 6.90 (s, 1H, CH^{β}).

It therefore appears that ^1H NMR shifts of imidazole CH protons are surprisingly insensitive to tautomer state, and hence do not vary significantly upon titration of **1** with chloride. The relatively small response of ^1H NMR shifts to the substitution pattern of the two nitrogen atoms is also evident in reported ^1H NMR chemical shift values for the CH_{imid} protons in *N*-methylated imidazoles: the imidazole proton equivalent to our ' CH^{β} ' varies by 0.22 ppm, but ' CH^{β} ' is insensitive to a regioisomeric switch ($\Delta\delta$ 0.02)

K. Asuno, S. Matsubara
Org. Lett. 2010, 12 (21), 4988-4991

E. van den Berge, R. Robiette
J. Org. Chem. 2013, 78 (23), 12220-12223

It is for this reason that we focussed on the more diagnostic ^{13}C NMR values. An explanatory sentence clarifying our use of ^{13}C NMR data and the lack of information obtainable from ^1H shifts has been added to the paragraphs at the bottom of p3/top of p4 of the manuscript.

2) Chemical shifts in Figure S113 are clearly describing a process more complicated than a 1:1 stoichiometry. From dimerization data it does seem related to aggregation processes. However, chemical shift variations are not linear (e.g. NH_4 going first up and then down) indicating a more complex recognition pattern. This should be clarified (see also errors in the binding fitting).

We agree that some features of the NMR data for **1** suggest more complex behaviour than 1:1 binding, but it is worth noting the context of these more unusual features, before attempting to draw detailed conclusions. The principal changes in chemical shift take place at the binding site protons NH_1 and NH_2 (changing by 0.9-1 ppm). Changes in the chemical shift of NH_4 and 5 are interesting, but are an order of magnitude smaller (Fig S13b). These changes are not due to binding of chloride to NH_4 , as this would lead to a downfield (rather than an upfield) shift, and no equivalent shift is seen in compounds lacking the disubstituted BTMP urea binding site (SI Section 5.4. Figures S55-56).

The small changes in NH_4 and NH_5 are also unusually subject to a higher degree of uncertainty because these signals are broadened in **1**, even in the absence of chloride. There are also signal overlaps with other broadened peaks of compound **1** (see Figure S1, NH_2 , 3, 4, 5 and corresponding peaks in Fig S12). The lack of precision in determining the small chemical shift changes for these protons may be part of the reason that a variation of this type is not evident in the inner urea NH signals within either of the simpler foldamers **2** (SI Section 5.2.3, Figure S32) or **3** (Section 5.3.3, Figures S41-S43) on titration with tetrabutylammonium chloride. We are therefore very wary of using these small variations to draw any detailed conclusions.

However, we can propose some possible explanations. A small degree of self-aggregation of compound **1** does indeed occur at room temperature as shown in Figure S19 ($K_{\text{assoc}} = 31 \pm 2 \text{ M}^{-1}$). Small amounts of higher order aggregates are possible at the start of the titration, especially as head-to-tail interactions have been observed in related structures (*Chem. Sci.* **2022**, **13**, 13153-13159): these aggregates would feature hydrogen bond directionalities opposite to that of the major conformer. The initial addition of ligand would disrupt these aggregates, returning more of the foldamer to its 'native' hydrogen bond directionality, before coordination of the ligand at the binding site dominates as the ligand concentration increases. These competing equilibria during the titration could explain the non-linear effects.

To clarify the fact that these effects exist, but that they are small and do not have a clear explanation, we have added clarifying text on p7 of the manuscript to point out that although weak higher order complexation cannot be excluded with these data (Figure S15), it does not affect our conclusions, and that weak higher-order aggregation of **1** may be disrupted by addition of chloride (SI Section 5.1.2 and Figure S9, S10).

3) How the authors are sure that the halide is not binding within the foldamer amides and only in the external part? This point needs also to be clarified.

We were very keen to establish that the effects we observe result from an interaction at the N,N'-disubstituted urea binding site and are not due to chloride binding to the trisubstituted ureas closer to the imidazole. We therefore carried out the control experiments described in SI Section 5.4 (Figures S56-57). These show that titration of model foldamers containing trisubstituted ureas but no N,N'-disubstituted urea binding sites lead to minimal changes in chemical shift, indicating very weak interaction between chloride and the urea chain. These experiments were referred to in the sentence 'Control experiments on oligoureas **SI5** and **SI6** ... ruled out the possibility that the chemical shift variations measured during the titration arise from a direct interaction between Bu₄NCl and the hydrogen-bond chain (Figures S56 and S57).', but we have now carried out more work to further confirm this lack of direct interaction with the internal ureas. Titration of compound **2** (with an imidazole but without a disubstituted urea binding site) with tetrabutylammonium chloride (SI Section 5.2.3., Figure S32) showed no binding-induced shifts at inner the urea NH protons (<0.03 ppm), so this information has also been added to the manuscript (text on p7).

Reviewer 3

(1) Synthesis of ethylene-bridged oligoureas: it is not clear to me whether the synthesis scheme presented in the SI includes common intermediates with analogues published in refs 36-38. I wonder whether 1c, for example, is new or already used in previous studies.

Compounds **1c**, **3a-c** and **SI6** have been reported previously. We have accordingly amended the caption to scheme S3, added references to the literature describing the preparation of compounds **1c**, **3a-c** and **SI6**, and removed the description of their synthesis and characterisation.

(2) Tautomerization rate: population of imidazole tautomers lead to two separate peaks in 13C NMR instead of four which means that the exchange is rapid on the 13C NMR timescale. Is it possible to have a more quantitative assessment of the ratio of 4- and 5-isomers?

Tautomer exchange is likely to be very fast, and very low temperature NMR experiments with these foldamers are hampered by precipitation, so we were unable to quantify tautomer ratios directly. At room temperature all NMR spectra show weighted averages of the two tautomers, but in the absence of data on the slow exchange chemical shift values, it is not possible to determine ratios with accuracy. Nonetheless, the change in the diagnostic ¹³C separation from a value close to that calculated for one tautomer (and experimentally observed for an equivalent regioisomer) to one closer to that calculated (and equivalently observed for the other regioisomer) for the other is strongly indicative of a switch in tautomer preference, though probably not with complete selectivity. This section of the manuscript (p4) has now been reworded, and Figure 2 reorganised, to clarify these observations and the degree of confidence we have in interpreting them.

(3) Probing of the bonding of chloride: interpretation of the N–H shifts of 1 upon chloride binding are not obvious.

Why NHC appears upfield shifted?

Our interpretation of this observation, shown in the titration shown in Figure 4b, is that NHc is initially hydrogen-bonded to the carbonyl of a more electron-rich N-alkyl urea; on reversal of directionality, it is still hydrogen bonded, but to a more electron-deficient N-aryl urea. The strength of this new hydrogen bond is likely to be weaker, thus the chemical shift of this NH proton moves upfield.

Why the imidazole N-H is not mentioned as potential probe to follow this binding?

The ¹H NMR chemical shift of imidazole NH would be very diagnostic of hydrogen bonding and tautomer state, but unfortunately it is never visible in the NMR spectra, even at low temperature. We assume this is due to exchange broadening. A sentence has been added to the manuscript on p7 to explain this.

FT-IR would have been a particularly suitable to follow this experiment in solution.

This suggestion is very interesting as FT-IR will always show slow exchange between tautomers and conformers. However, the identification of each NH (or each C=O) among several in each molecule would make interpretation of the results very challenging. We did consider carrying out an additional experiment of this type, but unfortunately the greater quantities of material required for these IR studies were unavailable.

The authors should also consider that other stoichiometries are possible (2:2, etc..) and that minor species with chloride at other positions cannot be excluded.

Please see our response to referee 2 for a discussion of the existence and importance of these effects. The next text on p7 clarifies the fact that such effects are weak, although there is some evidence that they happen.

(4) Difference in reactivity between 2 and 2+chloride: The small effect of chloride on the activity of 2, which makes 1+Cl and 2+Cl of similar reactivity, is explained by "increase in solvent polarity". However, such an effect is not exhibited by Bu₄NCl alone. The authors must precise their interpretation.

The reactivity in question is the rate of acylation by 4 of the imidazole terminus of 1 or 2 in the presence of Bu₄NCl. An experiment with Bu₄NCl and 4 alone does not give a product because there is no imidazole nucleophile to react, so no observation can be made.

Finally, a more precise background of the work will be given by adding existing literature on the effect of chloride on tuning the selectivity of helical catalysts (*Chem. Commun.*, 2019, 2162) as well as quoting references dealing with switchable catalysts from the Schmittel group.

As suggested, we have added the following references to the manuscript:

Li, Y., Caumes, X., Raynal, M. & Bouteiller, L. Modulation of catalyst enantioselectivity through reversible assembly of supramolecular helices. *Chem. Commun.* **55**, 2162-2165 (2019).

Schmittel, M. **Switchable Catalysis Using Allosteric Effects**, in *Supramolecular Catalysis: New Directions and Developments*, Eds. Matthieu Raynal and Piet van Leeuwen, Wiley **2022**, Chapter 39, p. 575-589. ISBN:9783527349029.

De, S., Pramanik, S. & Schmittel, M. A Toggle Nanoswitch Alternately Controlling Two Catalytic Reactions. *Angew. Chem., Int. Ed.* **53**, 14255–14259 (2014).

Biswas, P. K., Saha, S., Gaikwad, S. & Schmittel, M. Reversible Multicomponent AND Gate Triggered by Stoichiometric Chemical Pulses Commands the Self-Assembly and Actuation of Catalytic Machinery. *J. Am. Chem. Soc.* **142**, 7889–7897 (2020).

I also encourage the authors to organize better their SI file (130 pages) which is quite hard to follow.

In order to simplify the Supporting Information file, we have removed descriptions associated with published compounds, and we have added further subheadings in order to assist with navigation. We have also improved the pagination of the file, and changed the order of some of the paragraphs in order to provide a more coherent narrative. We have extended the table of contents and added hypertext links to the table of contents to assist with navigation.

Reviewers' Comments:

Reviewer #1:

Remarks to the Author:

Tilly and coauthors made significant additions to the paper with K_a and rate data for competing anions, Br^- , I^- , and NO_3^- . The new experiments directly address the question of selective activation by Cl^- and strengthens the work overall. Chloride produces the largest rate enhancement, while having a moderate K_a compared to nitrate. This indicates the rate enhancement is dependent on the anion and conformation changes caused by binding of the anion to the oligomer. The new experiments further support the enzyme-like properties of this system with some selectivity for the bound molecule. The additional experiments are supported by data included in the SI and archived in an open database. The authors conclusions are well supported by the available data.

There is evidence for some complex binding behavior beyond 1:1, which the authors addressed with control and dimerization studies. Overall, the additional binding constants appear to be weak and not directly attributed to the observed reaction activation. The reference below may be a helpful addition for the complex behavior of TBA salts in organic solvents.

1. Liu, Y., Sengupta, A., Raghavachari, K. & Flood, A. H. Anion Binding in Solution: Beyond the Electrostatic Regime. *Chem* 3, 411–427 (2017).

Reviewer #2:

Remarks to the Author:

The authors have exhaustively answered to all the questions. In my opinion, the paper can be published without other revisions.

Reviewer #3:

Remarks to the Author:

By including a more detailed discussion about the chemical shift variations induced by the complexation of chloride anion on foldamer 1, as well as about by detailing the diagnostic signals for probing tautomeric ratios, the authors have clarified important points that make the manuscript suitable for publication in *Nature Communications*.

Thank you for your message regarding this paper. We have included the citation to the reference suggested by reviewer 1.